# Library Learning Doesn't: The Curious Case of the Single-Use "Library"

**Ian Berlot-Attwell**
University of Toronto
Vector Institute
ianberlot@cs.toronto.edu

**Frank Rudzicz**
Dalhousie University
Vector Institute
frank@dal.ca

**Xujie Si**
University of Toronto
Vector Institute
six@cs.toronto.edu

## Abstract

Advances in Large Language Models (LLMs) have spurred a wave of LLM library learning systems for mathematical reasoning. These systems aim to learn a reusable library of *tools*, such as formal Isabelle lemmas [Paulson, 1994] or Python programs that are tailored to a family of tasks. Many of these systems are inspired by the human structuring of knowledge into reusable and extendable concepts [Ellis et al., 2021], but do current methods actually learn reusable libraries of tools?

We study two library learning systems for mathematics which both reported increased accuracy: LEGO-Prover [Wang et al., 2024a] and TroVE [Wang et al., 2024b]. We find that function reuse is extremely infrequent on miniF2F [Zheng et al., 2022] and MATH [Hendrycks et al., 2021]. Our followup ablation experiments suggest that, rather than reuse, self-correction and self-consistency are the primary drivers of the observed performance gains. Our code and data are available at https://github.com/ikb-a/curious-case.

## 1 Introduction

Mathematical progress is made by building with, and building upon, the tools of those who came before. Consequently, it is no surprise that there is research interest in developing systems that can automatically learn such reusable mathematical tools. Recently, LLMs have enabled new tool-learning methods with improved performance [Wang et al., 2024a,b, Zhang et al., 2024a, Yuan et al., 2024] – but are these systems truly learning generalized, reusable knowledge or is performance improved through other mechanisms? In this work, we study two prior systems: LEGO-Prover which aims to learn reusable formal Isabelle lemmas, and TroVE which aims to learn reusable Python functions. For both, our analysis of the model's behaviour reveals that direct reuse is negligible. Furthermore, we perform two ablation studies supporting our position that function reuse plays a limited role in these systems' improved mathematical reasoning.

## 2 Related Work

LLM library learning, i.e., creating and reusing tools, depends on LLMs' ability to use tools. Prior evaluations of tool-use (typically assuming tools as REST APIs) [Qu et al., 2024] included real-world queries [Yan et al., 2024], dedicated test environments [Li et al., 2023], and metrics ranging from LLM-as-a-judge [Guo et al., 2024] to tracking task-checkpoint completion [Lu et al., 2024].

38th Conference on Neural Information Processing Systems (NeurIPS 2024).

Table 1: Lemma reuse in LEGO-Prover released logs. Note that **lemma reuse is very uncommon**, and **no lemma reused twice**. For each split, we report the number of problems solved, the number of unique lemmas occurring in the PROVER's input prompts, the number of lemmas reused verbatim once, or more than once, and the number of lemmas whose *name* is reused once, or more than once. A lemma is reused $N$ times if it appears in $N + 1$ solutions (i.e., the initial use, and then $N$ reuses).

| | | | Verbatim reused | | Name reused | |
|---|---|---|---|---|---|---|
| Split | Problems Solved | Lemmas in Prompts | 1 | 2+ | 1 | 2+ |
| valid+GPT | 127 | 374 | 0 | 0 | 1 | 0 |
| valid+Human | 135 | 265 | 0 | 0 | 1 | 0 |
| test+GPT | 111 | 255 | 0 | 0 | 2 | 0 |
| test+Human | 122 | 339 | 1 | 0 | 2 | 0 |

In contrast, the evaluation of library learning systems has been limited. Accuracy is the metric of choice [Wang et al., 2024a,b, Zhang et al., 2024a, Yuan et al., 2024], but cannot capture the extent or quality of reuse: an excellent library is useless to a weak reasoner, and a powerful reasoner can ignore a useless library and derive results from first principles. Prior attempts to evaluate library learning have been limited to static measures of individual functions such as cyclomatic complexity [McCabe, 1976, Zhang et al., 2024a] and abstract syntax tree depth [Wang et al., 2024b], or have answered specific questions such as the ease of human verification [Wang et al., 2024b], accuracy under domain transfer [Zhang et al., 2024a, Qian et al., 2023], or performance in the sub-problem of refactoring ground truth solutions[Lin et al., 2024].

In this study, we evaluate two library learning systems for mathematical reasoning: LEGO-Prover, and TroVE (see Sections 2.1 and 2.2). For a review of library learning systems, see Appendix A.

## 2.1 LEGO-Prover: Purpose & Architecture

LEGO-Prover consumes a set of proposed theorems to produce corresponding formal Isabelle [Paulson, 1994] proofs. It was evaluated on the miniF2F [Zheng et al., 2022] dataset: each problem was attempted 100 times, and the system obtained feedback from the Isabelle verifier after each attempt. LEGO-Prover was designed to perform library learning. Using the term *skills* in place of *tools*, Wang et al. [2024a] claimed that "LEGO-Prover enables LLMs to utilize existing skills retrieved from the library" and "[m]odular and reusable skills are constantly added to the library to enable tackling increasingly intricate mathematical problems." LEGO-Prover performs library learning via two LLM systems: 1) The PROVER which uses the library to create proofs, and 2) the EVOLVER which iteratively refines the library. They communicate through shared databases, such as the *request db* which stores proposed lemmas to be proven and added to library.

## 2.2 TroVE: Purpose & Architecture

TroVE is a "method for inducing a toolbox of reusable functions to use in solving programmatic tasks," designed to receive a stream of word problems without a ground truth or verifier [Wang et al., 2024b]. For each problem, it attempts to produce a Python program that prints the correct solution. TroVE's mathematical reasoning was evaluated with the MATH dataset Hendrycks et al. [2021]. Each problem is considered once: an LLM generates 15 solutions, and the best is selected based on self-consistency (i.e., majority vote) [Wang et al., 2023]. In generation, 5 solutions ignore the library and directly generate a program (SKIP mode), 5 create a reusable helper function for inclusion in the library (CREATE mode), and 5 use a function from the library (IMPORT mode).

## 3 Analysis of LEGO-Prover

We begin by analyzing the publicly released LEGO-Prover evaluation log files [1] [Wang et al., 2024a]. These logs are a subset of the unreleased PROVER logs corresponding to the final attempts on the

---

[1]`https://github.com/wiio12/LEGO-Prover/blob/357672c7751cd0c84aff6bf72a3d1bf97614e81d/result/lego_result.zip`

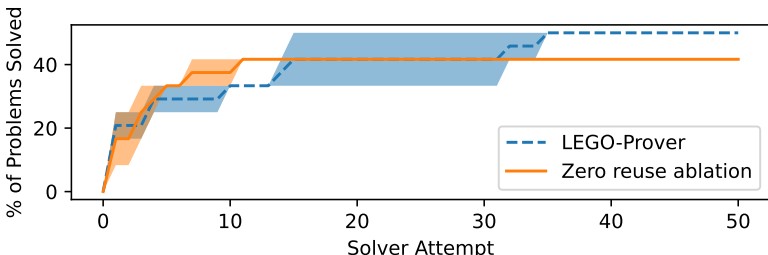

Figure 1: LEGO-Prover performance on a subset of the miniF2F validation split. The ablated model cannot reuse lemmas and performs similarly. The shaded region is one standard deviation, capturing variations in LLM output and race conditions.

successfully solved problems. Note that LEGO-Prover was evaluated on 4 data splits, and learned over 20,000 lemmas overall [Wang et al., 2024a].

We find that only 1,233 lemmas (∼6%) are used in the final solving step (i.e., are inputs to the PROVER). Of these, exactly one lemma is reused by the PROVER, and it is reused once (i.e., appears verbatim in two solutions). As the PROVER may be adjusting a lemma (e.g., paraphrasing, commenting, etc...) we repeat the analysis, checking only for the lemma's name. Again, lemma reuse is rare, and no lemma is reused more than once (i.e., no lemma has its *name* appear in 3 or more solutions). See Table 1 for details. For an example of verbatim vs. name use, see Appendix B.

Given these findings, there are only two possibilities by which LEGO-Prover may be performing reuse: 1) indirect reuse (e.g., the learned tools are useful, reusable exemplars, rather than directly used in the final solution), or 2) direct reuse occurs in the EVOLVER.

Instead, we hypothesize that reuse is not significantly boosting performance. We propose that self-correction [Pan et al., 2023] via the *request db* is the main mechanism of action. Note that the PROVER populates the *request db* by: 1) adding lemmas that the LLM suggests may be helpful sub-steps, and 2) adding lemmas from solution attempts that Isabelle could not verify. The EVOLVER uses the *request db* to modify existing tools to "aid in solving requests", and to "resolv[e] decomposed sub-goals" using the library [Wang et al., 2024a]. Thus, the performance gains may be due to a combination of chain-of-thought [Wei et al., 2022] (through the PROVER's proposal of helpful lemmas for the EVOLVER to solve) and self-correction (through the EVOLVER's retrying of failed lemmas).

To test whether any form of reuse is increasing performance, we ablate LEGO-Prover to remove cross-problem sharing: each theorem is solved with its own independent state and databases. E.g., in place of a global *request db*, each problem now has its own independent *request db*. We evaluate on a random size 12 subset of the validation split and use 50 attempts per problem. We perform our ablation using OpenAI's GPT-4o-mini as the original results were published using now deprecated versions of GPT-3.5-Turbo; see Appendix E for full details of the ablation. Running 2 trials, we find that the ablation's performance is strong, solving only 1 question less than the baseline (see Figure 1). Studying the problems solved by only the baseline, we find that only the simplest of the input lemmas are possibly used (namely $a^2 \geq 0$ and $ax^2 + bx + c = 0 \Rightarrow c = -(ax^2 + bx)$; see Appendix C). It is unclear as these facts are not treated as lemmas, and are given different justifications. This suggests that: 1) the LLM may be too weak if it needs examples of basic facts 2) the LLM struggles at reuse as it does not copy the given, verified, proofs.

## 4 Analysis of TroVE

As TroVE logs were not released, we re-ran TroVE on MATH, achieving accuracy within ±2% (absolute) of reported (see Appendix, Table 3). Note that the TroVE library also learns import statements; we ignore these in our analysis for two reasons. Firstly, our interest is in whether the system learns and reuses non-trivial tools, unlike statements such as "`import math`" and "`from sympy import symbols`". Secondly, as TroVE includes the entire library as part of the IMPORT prompt, and import statements are innately simple, it is impossible to determine whether an import statement is included in the LLM output due to reuse, or the LLM's innate knowledge.

Table 2: TroVE performance on MATH for the ablation and the baseline. Mean and standard deviation over 5 trials are reported. The variations arise from LLM output. † indicates that mean ablation performance is significantly strictly higher than the baseline's, at the Bonferroni-corrected 0.05 level, using a 2-sample 1-sided Welch's t-test (note, this test assumes approximate normality).

| | Accuracy on MATH test split | | | |
|---|---|---|---|---|
| Model | count | geo | inte | num |
| TroVE Reproduced | $0.236 \pm 0.008$ | $\mathbf{0.058} \pm 0.004$ | $0.120 \pm 0.006$ | $0.258 \pm 0.007$ |
| No Reuse Ablation | $\mathbf{0.250} \pm 0.000$† | $0.050 \pm 0.000$ | $\mathbf{0.134} \pm 0.014$ | $\mathbf{0.290} \pm 0.014$† |

Analyzing the logs, we find that TroVE's final libraries only contain 15 learned functions, having learned functions for only 3 of the 7 MATH subject test splits: counting, number, and pre-algebra. No functions are learned in the algebra, geometry, intermediate algebra, or pre-calculus splits. Of the 15 learned functions, only 2 are reused in a correct solution: `is_perfect_square(n)` is reused in one correct solution and `is_prime(num)` is reused in two correct solutions.

Given 3 successful reuses in 3,201 test questions, we believe that TroVE's improvements over the baselines are not due to function reuse. Instead, we believe that ensembling and self-consistency are responsible. To test this, we ablate the model by disabling IMPORT mode, but maintaining the 15 solution attempts: we generate 8 solutions ignoring the library (i.e., SKIP mode) and 7 attempting to create a helper function (i.e., CREATE mode). As in the original work we use `CodeLlama-7b-Instruct-hf` [Rozière et al., 2023]; see Appendix F for the full ablation details. Ablating IMPORT mode prevents reuse as the library never appears in the model's input, thus also preventing library learning of import statements. As to why this ablation could still be performant, prior work established the benefits of self-consistency and increased sampling [Brown et al., 2024], and it's known that library-less tool-creation can boost performance by forcing abstract reasoning [Yuan et al., 2024].

We evaluate our ablated model on the intermediate_algebra test split (reportedly the largest performance gain over non-reuse baselines), and the geometry, number, and count test splits. On the intermediate_algebra, number, and count splits, our ablation exceeds the baseline's performance, with the improvement being statistically significant on two splits (See Table 2). On only the geometry split does the base model perform slightly better, though the learned libraries only contains import statements. From this we can conclude that library learning *import statements* can be slightly beneficial, but only for certain domains. Typically, TroVE's library learning degrades its performance.

## 5 Conclusions

In this study, we find that both TroVE and LEGO-Prover do not directly reuse the tools they learn. Furthermore, the results of our ablations suggest that their performance gains cannot be solely attributed to indirect reuse either.

We intend that this paper be a call for the better understanding of the limitations of current library learning systems, and for improved evaluation. We show that accuracy is misleading in isolation: the system's reuse behaviour is paramount, and careful ablation is critical. Both papers studied made sensible claims as the created systems were deliberately designed for library learning and were tested against ablations that were not unreasonable – however they also relied heavily on accuracy as a metric instead of directly observing the systems' use of the library, and both chose ablations that in hindsight were too aggressive. It is clear that, particularly for ablations of library learning systems, minimal changes are preferable, and considerable thought should be put into other possible causes of improvements. There is a clear need for a broadly applicable framework for the evaluation of library learning specifically; this framework must rely on more than task accuracy and ablations to evaluate library learning and reuse.

Finally, considering library learning for mathematics in general: are LLMs capable learning tools and performing direct, verbatim reuse? Given that the observed improvements do not come from direct reuse, would direct reuse actually improve systems for mathematical reasoning, or is it overly brittle making soft reuse desirable? These important questions follow from our findings, and should inform the design of future research into library learning systems.

# 6    Limitations & Broader Impact

Due to resource constraints, our ablation studies could be more thorough. Most obviously, we only study two models, and on two datasets. The LEGO-Prover ablation is not ideal, as library learning is disadvantaged by operating on a subset of the questions; this was necessary due to resource constraints. Another limitation is that LEGO-Prover's databases are pre-loaded with the full dataset of problems; consequently, the EVOLVERs are exposed to other problem statements – note, however, that the impact on testing reuse is minimal. Firstly, the PROVER cannot attempt to solve any of these other problems, thus the *request db* cannot gain pending lemmas related to other problems. Secondly, under the ablated model, tasks cannot share lemmas – any performance gains would come from having access to other sample problems instead of reuse.

While we demonstrate that the performance gains in mathematical reasoning seen by TroVE and LEGO-Prover cannot be attributed to the direct learning and reuse of tools, there is a very important but *subtly different* question which remains unanswered: whether these systems are at all capable of library learning. It is possible that these systems have the capacity to learn reusable functions and lemmas, but the datasets do not provide the opportunity. Manually inspecting the MATH dataset, our tentative conclusion is that the dataset is intrinsically not amenable to function learning with Python – we suspect the questions are too diverse, with the shared components already being captured by standard libraries. How this could be more formally demonstrated remains an important open question that is beyond the scope of this work.

This work has no immediate societal impact, rather, it highlights current limitations and challenges assumptions in this field. However, deploying tool-learning systems may carry a security risk from executing LLM-generated code (we sandboxed TroVE). More generally, library learning systems are self-improving through code generation, an approach that has raised concerns [Zelikman et al., 2023]. Unexpected behaviours may develop, thus requiring sandboxing and monitoring, at the very least.

## Acknowledgments and Disclosure of Funding

Resources used in preparing this research were provided, in part, by the Province of Ontario, the Government of Canada through CIFAR, and companies sponsoring the Vector Institute `www.vectorinstitute.ai/partnerships/`. Generous support was also provided by the Microsoft Accelerating Foundation Models Research (AFMR) program.

We would also like to thank Zhiruo Wang, Zhaoyu Li, William Cunningham, and our anonymous reviewers for their time and conversations that helped in various ways to shape and improve this work. Finally, the lead author would like to thank Frank Rudzicz for years of guidance and support, and Xujie Si for both encouraging this work as being of interest to the mathematical reasoning community, and for providing critical resources without which it could not have been possible. Thank you everyone for helping make this work possible.

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

# Appendix

## A  Extended Related Work

Current LLM-based library learning systems tend to fall into two main camps: systems designed for general word problem solving, typically including mathematical reasoning and typically generating Python functions (e.g., Cai et al. [2024], Yuan et al. [2024], Wang et al. [2024b]), and agentic systems

designed to interact with a specific, complex environment (e.g., Wang et al. [2024c], Tan et al. [2024], Wu et al. [2024], Zhang et al. [2024a], Zhao et al. [2024]).

Generally, such systems access the library via in-context learning (ICL); some place the entire library in the context [Wang et al., 2024b, Zhang et al., 2024a], whereas others first use a semantic-similarity retrieval step to allow for larger libraries. Yuan et al. [2024] in particular uses a retrieval system that incorporates a LLM-generated description of the tool to be retrieved; LEGO-Prover behaves similarly by having several phases where the system alternates between proposing useful tools to be added to the library, attempting to create these tools, and possibly retrieving these tools.

These systems are typically bottom-up (iteratively developing a library over time), though a handful of top-down approaches exist. These top-down approaches instead decompose a high-level description of the tasks into reusable modules [Chen et al., 2024, Kuan et al., 2024, Zhao et al., 2024, Zhang et al., 2024b]; to the best of the authors' knowledge this approach is yet to be applied to mathematical reasoning.

These LLM-based systems typically attempt to produce reusable tools via ICL: prompting the LLM to generate "reusable functions". In comparison, an older family of library learning work (e.g., Dreamcoder [Ellis et al., 2021] and LILO [Grand et al., 2024]) instead frame library learning as a matter of compression. In principle a function that compresses a set of solutions must be broadly applicable, and in practice a high-level function reduces the symbolic search space for program induction. More generally, compression has been of long standing interest in the field of artificial intelligence. Rendell [1983] defined conceptual knowledge as the ability to compress a raw space of possibilities into useful classes, and there are long-standing connections between compression and inductive reasoning. Framing inductive reasoning as the task of capturing the underlying pattern in a provided substring for the purposes of prediction, Solomonoff [1964] formalized induction as Bayesian reasoning under a prior favouring low Kolmogorov complexity. In other words, formalizing the concept of Occam's razor – that the simplest solution, that which can be highly compressed into a short description, is more likely. For a recent treatise on the value of compression, specifically within the area of mathematical reasoning, see Bengio and Malkin [2024].

Turing our attention to mathematics, deep learning in general and LLMs in particular have found broad application in theorem proving [Li et al., 2024]. Considering library learning specifically, a very closely related branch of work considers the problem of refactoring a collection of ground-truth solutions into reusable components. ATG [Lin et al., 2024] and REFACTOR [Zhou et al., 2024] train models to extract reusable formal lemmas from a provided set of ground-truth formal proofs. Similarly, ReGAL [Stengel-Eskin et al., 2024] refactors ground-truth Python solutions for the MATH dataset into a reusable library. These systems are valuable and may represent a better first step towards reusable knowledge, but their dependence on ground-truth solutions prevents them from being conventional library learning systems. In comparison, LEGO-Prover attempts to learn reusable lemmas and produce formal proofs from only formal problem statements, and informal natural language proofs – furthermore, Wang et al. [2024a] demonstrated that the latter could be automatically generated by ChatGPT with only a small degradation in system performance.

## B   Example of Verbatim Use versus Name Use by LEGO-Prover

Figure 2 is an example of verbatim use where an input lemma to the PROVER is used verbatim in the outputted solution.

In contrast, Figure 3 is an example of name use, where the name of the input lemma appears in the solution. In this case, the contents of the lemma are similar, but have significant differences. Note that an instance of verbatim use would, necessarily, also be an instance of name use.

A lemma is reused $N$ times if it is used $N + 1$ times – i.e., if the lemma is used in $N + 1$ solutions.

## C   LEGO-Prover Solutions not Found by Reuse-Free Ablation

We performed two runs of the original model, in both cases it outperformed the ablation by solving one additional problem. We present the found proofs and input lemmas in Figures 4 and 6. For improved legibility, we also provide a typeset approximation in Figures 5 and 7. In addition to the

| PROVER input | PROVER output |
|---|---|
| ```[... System prompt ...]``` 

 ```Here some useful skill for reference:``` 
 ```###### useful skill 1: ######``` 
 ` ```isabelle ` 
 **lemma step3:** 
   **fixes x y ::real** 
   **shows "x^2 + y^2 + (x^2 \* y^2) + 1 \\<ge> 1"** 
 **proof -** 
   **have "x^2 + y^2 + (x^2 \* y^2) + 1 - 1 = x^2 + y^2 + (x^2 \* y^2)"** 
     **by simp** 
   **then have "x^2 + y^2 + (x^2 \* y^2) + 1 - 1 \\<ge> 0"** 
     **by simp** 
   **then show ?thesis** 
     **by simp** 
 **qed** 
 ` ``` ` 

 ```###### useful skill 2: ######``` 
 ` ```isabelle ` 
 ```lemma algebra_sqineq_2xyxysq:``` 
 ```  fixes x y :: real``` 
 ```  shows "x^2 + y^2 \<ge> 2 * x * y"``` 
 ```proof -``` 
 ```  have "(x - y)^2 \<ge> 0"``` 
 ```    by simp``` 
 ```  then have "x^2 - 2 * x * y + y^2 \<ge> 0"``` 
 ```    by (simp add: algebra_simps power2_diff)``` 
 ```  then have "x^2 + y^2 \<ge> 2 * x * y"``` 
 ```    by simp``` 
 ```  then show ?thesis``` 
 ```    by simp``` 
 ```qed``` 
 ` ``` ` 

 ```[... Additional skills and example problems ...]``` 

 ```## Problems``` 
 ```What is the least possible value of $(xy-1)^2+``` 
 ```(x+y)^2$ for real numbers $x$ and $y$?``` 
 ```$\textbf{(A)} ~0\qquad\textbf{(B)} ~\frac{1}``` 
 ```{4}\qquad\textbf{(C)} ~\frac{1``` 
 ```}{2} \qquad\textbf{(D)} ~1 \qquad\textbf{(E)} Show``` 
 ```that it is \textbf{(D)} ~1.``` 

 ```## Informal proof``` 
 ```Step 1: Expand the expression $(xy-1)^2+(x+y)^2$``` 
 ```to obtain $x^2+2xy+y^2+x^2y^2-2xy+1$.``` 
 ```Step 2: Simplify the expression to``` 
 ```$x^2+y^2+x^2y^2+1$.``` 
 ```Step 3: Observe that all squares are nonnegative,``` 
 ```so the minimum value of the expression is 1.``` 
 ```Step 4: Show that the minimum value of 1 can be``` 
 ```achieved when $x=y=0$.``` 

 ```## Formal statement``` 
 ` ```isabelle ` 
 ```theorem amc12a_2021_p7:``` 
 ```  fixes x y ::real``` 
 ```  shows "1 \<le> ((x * y) - 1)^2 + (x + y)^2"``` 
 ```  apply (auto simp:algebra_simps power2_eq_square)``` 
 ` ``` ` 

 ```## Proof``` | ` ```isabelle ` 
 ```theory amc12a_2021_p7``` 
 ```  imports Complex_Main``` 
 ```begin``` 

 ```(* lemma 1. A lemma that shows the distribution``` 
 ```properties of real numbers will be helpful for``` 
 ```step 1, which requires this lemma to expand the``` 
 ```equation *)``` 
 ```lemma step1:``` 
 ```  fixes x y :: real``` 
 ```  shows "(x * y - 1)^2 + (x + y)^2 = x^2 + 2 * x *``` 
 ```y + y^2 + x^2 * y^2 - 2 * x *``` 
 ```  y + 1"``` 
 ```  by (simp add: power2_eq_square algebra_simps)``` 

 ```(* lemma 2. A lemma that shows the properties of``` 
 ```quadratic equation will be useful for step 2,``` 
 ```which requires this lemma to simplify the equation``` 
 ```*)``` 
 ```lemma step2:``` 
 ```  fixes x y ::real``` 
 ```  shows "x^2 + 2 * x * y + y^2 + x^2 * y^2 - 2 * x``` 
 ```* y + 1 = x^2 + y^2 + x^2 * y^2 + 1"``` 
 ```proof -``` 
 ```  have "x^2 + 2 * x * y + y^2 + (x^2 * y^2) - 2 *``` 
 ```x * y + 1 = x^2 + y^2 + (x^2 * y^2) + 1"``` 
 ```    by (simp add: algebra_simps)``` 
 ```  then show ?thesis``` 
 ```    by simp``` 
 ```qed``` 

 ```(* lemma 3. A lemma that shows the properties of``` 
 ```quadratic equation will be useful for step 3,``` 
 ```which requires this lemma to conclude the equation``` 
 ```*)``` 
 **lemma step3:** 
   **fixes x y ::real** 
   **shows "x^2 + y^2 + (x^2 \* y^2) + 1 \\<ge> 1"** 
 **proof -** 
   **have "x^2 + y^2 + (x^2 \* y^2) + 1 - 1 = x^2 + y^2 + (x^2 \* y^2)"** 
     **by simp** 
   **then have "x^2 + y^2 + (x^2 \* y^2) + 1 - 1 \\<ge> 0"** 
     **by simp** 
   **then show ?thesis** 
     **by simp** 
 **qed** 

 ```(* formal statement copy from the input *)``` 
 ```theorem amc12a_2021_p7:``` 
 ```  fixes x y ::real``` 
 ```  shows "1 \<le> ((x * y) - 1)^2 + (x + y)^2"``` 
 ```  apply (auto simp:algebra_simps power2_eq_square)``` 
 ```  using step1 step2 step3 by auto``` 

 ```end``` 
 ` ``` ` 

 ```This is the proof for the given problem.``` |

Figure 2: Example of verbatim reuse by the LEGO-Prover. The input lemma is reproduced exactly in the PROVER's output.

| PROVER input | PROVER output |
|---|---|
| [... System prompt ...]

Here some useful skill for reference:
###### useful skill 1: ######
```isabelle
theorem algebra_2rootspoly_apatapbeq2asqp2ab:
  fixes a b :: complex
  shows "(a+a) * (a+b) = 2 * a^2 + 2 * (a*b)"
proof -
  have "(a+a) * (a+b) = (a+a)*a + (a+a)*b"
    by (simp add: distrib_left)
  also have "... = 2*a^2 + 2*a*b"
    by (metis calculation distrib_left
mult.commute mult_2 numeral_2_eq_2
power2_eq_square)
  finally show ?thesis
    by simp
qed
```

###### useful skill 2: ######
```isabelle
lemma multi_distrib_complex:
  fixes a b c d :: complex
  shows "(a + b) * (c + d) = a * c + a * d + b * c
+ b * d"
  by (simp add: distrib_left distrib_right)
```

###### useful skill 3: ######
```isabelle
lemma algebra_1_generalized:
  fixes a b :: nat
  shows "a^2 + 2*a*b + b^2 = (a+b)^2"
  by (smt (z3) ab_semigroup_add_class.add_ac(1)
add.assoc add.commute add.left_commute power2_sum)
```

[... Additional skills and example problems ...]

## Problems
Simplify $(a-1)(a+1)(a+2) - (a-2)(a+1).$ Show that
it is a^3 + a^2.

## Informal proof
Step 1: Expand the expression $(a-1)(a+1)(a+2) -
(a-2)(a+1)$ by multiplying the binomials.
Step 2: Simplify the expression to obtain $(a^2-1)
(a+2)-(a-2)(a+1)$.
Step 3: Expand further to get $(a^3 + 2a^2 - a -
2) - (a^2 -a -2)$.
Step 4: Combine like terms to simplify the
expression to $a^3 + a^2$.
Step 5: Conclude that the simplified expression is
equal to $a^3 + a^2$.

## Formal statement
```isabelle
theorem mathd_algebra_568:
  fixes a :: real
  shows "(a-1) * (a+1) * (a+2) - (a-2) * (a+1) =
a^3 + a^2"
```

## Proof | ```isabelle
theory mathd_algebra_568
  imports Complex_Main
begin

(* Lemma 1: Distributive property of
multiplication to simplify the main equation *)
lemma algebra_1_generalized:
  fixes a b c :: real
  shows "(a - b) * (a + c) = a^2 + a * c - b * a -
b * c"
  by (simp add: algebra_simps)

(* formal statement copy from the input *)
theorem mathd_algebra_568:
  fixes a :: real
  shows "(a-1) * (a+1) * (a+2) - (a-2) * (a+1) =
a^3 + a^2"
proof -
  (* Step 1: Expand the expression $(a-1)(a+1)
(a+2) - (a-2)(a+1)$ by multiplying the binomials.
*)
  have c0: "(a-1) * (a+1) * (a+2) - (a-2) * (a+1)
= (a^2 - 1) * (a+2) - (a-2) * (a+1)"
    by (simp add: algebra_simps)

  (* Step 2: Simplify the expression to obtain $
(a^2-1)(a+2)-(a-2)(a+1)$. *)
  have c1: "(a^2 - 1) * (a+2) - (a-2) * (a+1) =
(a^3 + 2*a^2 - a - 2) - (a^2 - a - 2)"
    using algebra_1_generalized[of "a^2" 1 2]
algebra_1_generalized[of "a" 2 1]

    by (simp add: algebra_simps)

  (* Step 3: Expand further to get $(a^3 + 2a^2 -
a - 2) - (a^2 -a -2)$ *)
  have c2: "(a^3 + 2*a^2 - a - 2) - (a^2 - a - 2)
= a^3 + a^2"
    by simp

  (* Step 4: Combine like terms to simplify the
expression to $a^3 + a^2$. *)
  show ?thesis using c0 c1 c2
    by simp
qed

end
``` |

Figure 3: Example of name reuse by the LEGO-Prover. Only the name of the input lemma needs to be reproduced exactly in the output. In this case, the body of the input lemma has been significantly adjusted. Note Figure 2 is also an example of name reuse, as the input lemma's name appears in the solution (in that particular case, along with the rest of the lemma).

Table 3: TroVE performance on MATH. For comparison with Wang et al. [2024b], all reported numbers are best over 5 trials. Variation between trials arises from the stochastic sampling of the underlying LLM.

| | Best-of-5 accuracy on MATH test split | | | |
|---|---|---|---|---|
| Model | count | geo | inte | num |
| TroVE, Reported | **0.26** | **0.08** | 0.11 | 0.25 |
| TroVE Reproduced (ours) | 0.24 | 0.06 | 0.13 | 0.27 |
| TroVE, Reported CREATE-only ablation | 0.14 | 0.06 | 0.05 | 0.16 |
| No Reuse Ablation (ours) | 0.25 | 0.05 | **0.15** | **0.31** |

Table 4: LEGO-Prover hyperparameters

| Hyperparameter | value |
|---|---|
| Solution attempts per problem (num_attempts) | 50 |
| Number of PROVER processes (num_prover) | 3 |
| Number of EVOLVER processes (num_evolver) | 8 |
| Temperature (temperature) | 0.7 |

observations in the main paper, it should be noted that there is redundancy among the retrieved lemmas – deduplication and retrieval of lemmas remain areas for improvement.

# D  TroVE MATH reproduction

See table 3 for the best-of-five accuracies reported by TroVE, and achieved by our reproduction of their results.

# E  LEGO-Prover Hyperparameters and Experiment Details

At the time of publication, the LEGO-Prover logs released by Wang et al. [2024a] and used in our analysis are available at `https://github.com/wiio12/LEGO-Prover/blob/357672c7751cd0c84aff6bf72a3d1bf97614e81d/result/lego_result.zip`.

LEGO-Prover is built on OpenAI's GPT-3.5-Turbo and the 2022 release of the Isabelle proof assistant, specifically using its abilities as a proof verifier. Note that due to the deprecation of the LLMs originally used by LEGO-Prover (`gpt-3.5-turbo-0301`, `gpt-3.5-turbo-0613`, `gpt-3.5-turbo-16k`, `gpt-3.5-turbo-16k-0613`, `gpt-3.5-turbo-16k`, `gpt-3.5-turbo-16k-0613`), we upgrade the underlying LLM from GPT-3.5-Turbo to GPT-4o-mini.

We use the default LEGO-Prover hyperparameters, except for the number of retry attempts which, following Wang et al. [2024a]'s ablations, we reduce to 50. See Table 4 for details.

Note that the LEGO-Prover is initialized with a seed library of tools, and our ablation retains this initialization. The core claim we aim to disprove is that the model's performance gains predominantly come from reusable lemmas, and our ablation prevents any cross-task reuse.

The specific 12 problems chosen uniformly at random for our ablation study are: aime_1991_p6.json, algebra_2varlineareq_xpeeq7_2xpeeq3_eeq11_xeqn4.json, amc12a_2008_p15.json, amc12a_2013_p8.json, amc12a_2021_p7.json, amc12b_2002_p3.json, amc12b_2003_p9.json, mathd_algebra_31.json, mathd_algebra_109.json, mathd_algebra_116.json, mathd_numbertheory_149.json, and numbertheory_sqmod4in01d.json

Note that LEGO-Prover requires both the problem statement, and an informal natural language proof for conversion. We use the same human-generated informal proofs as Wang et al. [2024a]. The authors bundled said informal proofs inside of the miniF2F .json files listed above, available for download from `https://github.com/wiio12/LEGO-Prover/tree/`

| Input Lemmas | Final Proof |
|---|---|
| ```
###### useful skill 1: ######
lemma quadratic_root_substitution:
  fixes a b c k x :: real
  assumes "a * x^2 + b * x + c = 0"
  shows "c = - (a * x^2 + b * x)"
proof -
  obtain lhs where eq: "lhs = a * x^2 + b * x + c"
using assms by simp
  have "lhs = 0" using assms by (metis eq)
  thus ?thesis by (simp add: eq)
qed

###### useful skill 2: ######
lemma sqrt_limit_general:
  fixes x :: real
  assumes "n > 0" "k > 0" "k = sqrt(x + k)"
  shows "x = k^2 - k"
proof -
  have "k^2 = x + k" using assms(3) by (smt
(verit) assms(2) less_eq_real_def real_sqrt_le_iff
real_sqrt_pow2_iff real_sqrt_zero)
  then show ?thesis by auto
qed

###### useful skill 3: ######
lemma sqrt_difference:
  fixes a b :: real
  assumes "a >= 0" "b >= 0"
  shows "sqrt a - sqrt b = (a - b) / (sqrt a +
sqrt b)"
proof -
  have "sqrt a - sqrt b = (sqrt a + sqrt b) *
(sqrt a - sqrt b) / (sqrt a + sqrt b)"
    by (metis add.left_cancel
add_cancel_right_right add_le_same_cancel2
assms(1) assms(2)
cancel_comm_monoid_add_class.diff_cancel
dual_order.eq_iff eq_divide_eq mult.commute
nonzero_mult_div_cancel_right
real_sqrt_eq_zero_cancel_iff real_sqrt_ge_0_iff
real_sqrt_le_0_iff real_sqrt_zero)
  also have "... = (a - b) / (sqrt a + sqrt b)"
    by (smt (verit) assms(2) assms(2)
divide_cancel_right real_sqrt_le_mono
real_sqrt_less_mono real_sqrt_mult_self
square_diff_square_factored)
  finally show ?thesis
    by (simp add: field_simps)
qed

###### useful skill 4: ######
lemma sqrt_limit_generalized:
  fixes x :: real
  assumes "k = sqrt(x + k)" "k > 0"
  shows "x = k^2 - k"
proof -
  have "k^2 = x + k" using assms by (metis
add.commute real_sqrt_ge_0_iff real_sqrt_ge_zero
real_sqrt_gt_0_iff real_sqrt_pow2
verit_comp_simplify1(3) zero_le_mult_iff
zero_le_square)
  hence "x = k^2 - k" by simp
  thus ?thesis by auto
qed
``` | ```
theory mathd_algebra_116
  imports Complex_Main
begin
(* lemma 1: A lemma to calculate the square of a difference *)
lemma square_of_difference:
  fixes a b :: real
  shows "(a - b)^2 = a^2 - 2 * a * b + b^2"
proof -
  have "a - b = a + -b" by simp
  then show ?thesis by sos
qed

(* formal statement copy from the input *)
theorem mathd_algebra_116:
  fixes k x :: real
  assumes h0 : "x = (13 - sqrt 131) / 4"
    and h1 : "2 * x^2 - 13 * x + k = 0"
  shows "k = 19/4"
proof -
  (* Step 1: Substitute x into the quadratic equation to
express it in terms of k *)
  have "2 * x^2 - 13 * x + k = 0" using h1 by simp
  then have "k = 13 * x - 2 * x^2" by simp

  (* Step 2: Calculate 2 * x^2 *)
  have "x^2 = ((13 - sqrt 131) / 4)^2"
    using h0 by blast
  then have "x^2 = (13 - sqrt 131)^2 / 16"
    by (simp add: power2_eq_square)
  then have "2 * x^2 = 2 * (13 - sqrt 131)^2 / 16"
    by simp
  then have "2 * x^2 = (13 - sqrt 131)^2 / 8"
    by simp

  (* Step 3: Expand (13 - sqrt 131)^2 *)
  have "(13 - sqrt 131)^2 = 169 - 26 * sqrt 131 + 131"
    using square_of_difference[of 13 "sqrt 131"] by simp
  then have "(13 - sqrt 131)^2 = 300 - 26 * sqrt 131"
    by simp

  (* Step 4: Substitute back to find 2 * x^2 *)
  have "2 * x^2 = (300 - 26 * sqrt 131) / 8"
    by (metis \<open>(13 - sqrt 131)\<^sup>2 = 300 - 26 * sqrt
131\<close> \<open>2 * x\<^sup>2 = (13 - sqrt 131)\<^sup>2 /
8\<close>)

  (* Step 5: Calculate 13 * x *)
  have "13 * x = 13 * ((13 - sqrt 131) / 4)"
    using h0 by simp
  then have "13 * x = (169 - 13 * sqrt 131) / 4"
    by simp

  (* Step 6: Adjust the expression for k *)
  have "k = (169 - 13 * sqrt 131) / 4 - (300 - 26 * sqrt
131) / 8"
    using `k = 13 * x - 2 * x^2` by (metis \<open>13 * x =
(169 - 13 * sqrt 131) / 4\<close> \<open>2 * x\<^sup>2 = (300
- 26 * sqrt 131) / 8\<close> diff_add_cancel diff_diff_eq
diff_diff_eq2 diff_zero divide_numeral_1 eq_diff_eq
eq_iff_diff_eq_0 frac_eq_eq mult.commute mult.left_commute
mult_2_right mult_eq_0_iff mult_numeral_1_right
nonzero_mult_div_cancel_right power2_eq_square
right_diff_distrib' zero_neq_numeral)

  (* Step 7: Combine the fractions *)
  have "k = (169 - 13 * sqrt 131) / 4 - (300 - 26 * sqrt
131) / 8"
    by (metis \<open>k = (169 - 13 * sqrt 131) / 4 - (300 - 26
* sqrt 131) / 8\<close>)
  then have "k = (2 * (169 - 13 * sqrt 131) - (300 - 26 * sqrt
131)) / 8"
    by (simp add: field_simps)

  (* Step 8: Simplify the combined expression *)
  have "2 * (169 - 13 * sqrt 131) = 338 - 26 * sqrt 131"
    by simp
  then have "k = (338 - 26 * sqrt 131 - 300 + 26 * sqrt 131) /
8"
    by (smt (verit) \<open>k = (2 * (169 - 13 * sqrt 131) -
(300 - 26 * sqrt 131)) / 8\<close>)
  then have "k = (338 - 300) / 8" by simp
  thus ?thesis by simp
qed
end
``` |

Figure 4: LEGO-Prover input lemmas (left) and found proof (right). The proof proves that $\forall k \in \mathbb{R}$ : if $x = (13 - \sqrt{131})/4$ and $2x^2 - 13x + k = 0$ then $k = 19/4$. See Figure 5 for a typeset approximation, and commentary of LEGO-Prover's use (and failure to use) the input lemmas.

| Input Lemmas | Output Proof |
|---|---|
| Useful skill 1:
$\forall a, b, c, k, x \in \mathbb{R}:$
$ax^2 + bx + c = 0 \Rightarrow c = -(ax^2 + bx)$
Demonstrates: assms, simp, (metis eq), (simp add: eq) | Define Lemma square_of_difference:
$\forall a, b \in \mathbb{R}: (a-b)^2 = a^2 - 2ab + b^2$
Proof of Lemma:
$a - b = a + (-b)$
Method: simp
Lemma square_of_difference follows using method sos |
| Useful skill 2:
$\forall x \in \mathbb{R}: n > 0, k > 0$
$k = \sqrt{x+k} \Rightarrow x = k^2 - k$
Demonstrates: assms, (smt (verit) assms(2) less_eq_real_def real_sqrt_le_iff real_sqrt_pow2_iff real_sqrt_zero), auto | Define theorem mathd_algebra_116:
$\forall k \in \mathbb{R}:$
Assume $x = (13 - \sqrt{131})/4$
Assume $2x^2 - 13x + k = 0$
Then: $k = 19/4$
Proof:
$2x^2 - 13x + k = 0$
$k = 13x - 2x^2$
Method: simp |
| Useful skill 3:
$\forall a, b \in \mathbb{R}: \sqrt{a} - \sqrt{b} = (a - b)/(\sqrt{a} + \sqrt{b})$
Demonstrates: (metis add.left_cancel add_cancel_right_right add_le_same_cancel2 assms(1) assms(2) cancel_comm_monoid_add_class.diff_cancel dual_order.eq_iff eq_divide_eq mult.commute nonzero_mult_div_cancel_right real_sqrt_eq_zero_cancel_iff real_sqrt_ge_0_iff real_sqrt_le_0_iff real_sqrt_zero), (smt (verit) assms(1) assms(2) divide_cancel_right real_sqrt_le_mono real_sqrt_less_mono real_sqrt_mult_self square_diff_square_factored), (simp add: field_simps) | $x^2 = ((13 - \sqrt{131})/4)^2$
Method: blast

$x^2 = (13 - \sqrt{131})^2/16$
Method: (simp add: power2_eq_square)

$2x^2 = 2(13 - \sqrt{131})^2/16$
$2x^2 = (13 - \sqrt{131})^2/8$
Method: simp

$(13 - \sqrt{131})^2 = 169 - 26\sqrt{131} + 131$
Method: using lemma square_of_difference and sos

$(13 - \sqrt{131})^2 = 300 - 26\sqrt{131}$
Method: simp |
| Useful skill 4:
$\forall x \in \mathbb{R}: k = \sqrt{x+k}, k > 0 \Rightarrow x = k^2 - k$
Demonstrates: assms, (metis add.commute real_sqrt_ge_0_iff real_sqrt_ge_zero real_sqrt_gt_0_iff real_sqrt_pow2 verit_comp_simplify1(3) zero_le_mult_iff zero_le_square), simp, auto | $2x^2 = (300 - 26\sqrt{131})/8$
Method: (metis \<open>(13 - sqrt 131)\<^sup>2 = 300 - 26 * sqrt 131\<close>\<open>2 * x\<^sup>2 = (13 - sqrt 131)\<^sup>2 / 8\<close>)

$13x = 13((13 - \sqrt{131})/4)$
$13x = (169 - 13\sqrt{131})/4$
Method: simp

$k = (169 - 13\sqrt{131})/4 - (300 - 26\sqrt{131})/8$
Method: (metis \<open>13 * x = (169 - 13 * sqrt 131) / 4\<close>\<open>2 * x\<^sup>2 = (300 - 26 * sqrt 131) / 8\<close> diff_add_cancel diff_diff_eq diff_diff_eq2 diff_zero divide_numeral_1 eq_diff_eq eq_iff_diff_eq_0 frac_eq_eq mult.commute mult.left_commute mult_2_right mult_eq_0_iff mult_numeral_1_right nonzero_mult_div_cancel_right power2_eq_square right_diff_distrib' zero_neq_numeral)

$k = (169 - 13\sqrt{131})/4 - (300 - 26\sqrt{131})/8$
Method: (metis \<open>k = (169 - 13 * sqrt 131) / 4 - (300 - 26 * sqrt 131) / 8\<close>)

$k = (2(169 - 13\sqrt{131}) - (300 - 26\sqrt{131}))/8$
Method: (simp add: field_simps)
$2(169 - 13\sqrt{131}) = 338 - 26\sqrt{131}$
Method: simp

$k = (338 - 26\sqrt{131} - 300 + 26\sqrt{131})/8$
Method: (smt (verit) \<open>k = (2 * (169 - 13 * sqrt 131) - (300 - 26 * sqrt 131)) / 8\<close>)

$k = (338 - 300)/8$
Method: simp. Theorem follows. |

Figure 5: A typset approximation of LEGO-Prover input lemmas (left) and found proof (right). The proof proves that $\forall k \in \mathbb{R}$ : if $x = (13 - \sqrt{131})/4$ and $2x^2 - 13x + k = 0$ then $k = 19/4$. See Figure 4 for the original Isabelle lemmas and proof. Note that skill 1 may have been indirectly used in rewriting $2x^2 - 13x + k = 0$ as $k = 13x - 2x^2$. Skills 2, 3 and 4 do not seem to be used directly or indirectly; furthermore skills 2 and 4 are the same lemma. Their use of smt and metis may have encouraged the prover to use these same tools.

| Input Lemmas | Final Proof |
|---|---|
| Here some useful skill for reference:

###### useful skill 1: ######
lemma nonneg_square:
  fixes a :: real
  shows "a^2 \<ge> 0"
proof -
  have "a^2 = a * a" by (simp add:
power2_eq_square)
  thus ?thesis by (simp add: mult_nonneg_nonneg)
qed


###### useful skill 2: ######
lemma square_non_negative:
  fixes a :: real
  shows "a^2 >= 0"
proof -
  have "a^2 = a * a" by sos
  then show ?thesis by auto
qed


###### useful skill 3: ######
lemma square_sum_nonneg:
  fixes x y :: real
  shows "x^2 + y^2 \<ge> 0"
  by (simp)


###### useful skill 4: ######
lemma min_sum_of_squares:
  fixes x y :: real
  shows "x^2 + y^2 >= 0"
proof -
  have "x^2 >= 0" by auto
  moreover have "y^2 >= 0" by auto
  ultimately have "x^2 + y^2 >= 0 + 0" by (simp
add: add_nonneg_nonneg)
  thus "x^2 + y^2 >= 0" by simp
qed | theory amc12a_2021_p7
  imports Complex_Main
begin

(* Lemma: The expression can be expressed in terms
of squares to show its non-negativity. *)
lemma expression_nonneg:
  fixes x y :: real
  shows "((x * y) - 1)^2 + (x + y)^2 >= 1"
proof -
  (* Step 1: Define the expression we want to
minimize as \( E = (xy - 1)^2 + (x + y)^2 \). *)
  let ?E = "((x * y) - 1)^2 + (x + y)^2"

  (* Step 2: Expand the expression \( E \): \( (xy
- 1)^2 + (x + y)^2 \) *)
  have "((x * y) - 1)^2 = (x^2 * y^2 - 2 * x * y +
1)" by sos
  have "(x + y)^2 = x^2 + 2 * x * y + y^2" by sos

  (* This simplifies to: \( E = x^2y^2 + x^2 + y^2
+ 1 \) *)
  have "?E = (x^2 * y^2 - 2 * x * y + 1) + (x^2 +
2 * x * y + y^2)"
    by sos

  (* Thus, we have: \( E = x^2 * y^2 + x^2 + y^2 +
1 \) *)
  have "?E = x^2 * y^2 + x^2 + y^2 + 1" by sos

  (* Step 3: Recognize that \( x^2y^2 \), \( x^2
\), and \( y^2 \) are all non-negative. *)
  have "x^2 * y^2 >= 0" by auto
  have "x^2 >= 0" by auto
  have "y^2 >= 0" by auto

  (* Therefore, the minimum value of \( E \)
occurs when each of these terms is minimized. *)
  (* The minimum occurs when \( x = 0 \) and \( y
= 0 \). *)
  have "?E >= 1" by sos

  (* Step 4: Set \( x = 0 \) and \( y = 0 \) to
achieve the minimum: \( E(0, 0) = 1 \). *)
  have "?E = 1" when "x = 0" and "y = 0"
    using that by auto

  (* Step 5: Conclude that the least possible
value of \( E \) is \( 1 \). *)
  thus ?thesis by sos
qed

(* formal statement copy from the input *)
theorem amc12a_2021_p7:
  fixes x y ::real
  shows "1 \<le> ((x * y) - 1)^2 + (x + y)^2"
  apply (auto simp:algebra_simps power2_eq_square)
  using expression_nonneg by sos

end |

Figure 6: LEGO-Prover input lemmas (left) and found proof (right). The proof proves that $\forall x, y \in \mathbb{R} : 1 \le (xy - 1)^2 + (x + y)^2$. See Figure 7 for a typeset approximation, and commentary of LEGO-Prover's use (and failure to use) the input lemmas.

| Input Lemmas | Output Proof |
|---|---|
| Useful skill 1: $\forall a \in \mathbb{R} : a^2 \geq 0$
Demonstrates: (simp add: power2_eq_square), (simp add: mult_nonneg_nonneg)

Useful skill 2: $\forall a \in \mathbb{R} : a^2 \geq 0$
Demonstrates: sos, auto

Useful skill 3: $\forall x, y \in \mathbb{R} : x^2 + y^2 \geq 0$
Demonstrates: simp

Useful skill 4: $\forall x, y \in \mathbb{R} : x^2 + y^2 \geq 0$
Demonstrates: auto, (simp add: add_nonneg_nonneg), simp | Define Lemma expression_nonneg:
$\forall x, y \in \mathbb{R} : (xy - 1)^2 + (x + y)^2 \geq 1$
Proof of Lemma:
Let $E = (xy - 1)^2 + (x + y)^2$

$(xy - 1)^2 = (x^2 y^2 - 2xy + 1)$
$(x + y)^2 = x^2 + 2xy + y^2$
$E = (x^2 y^2 - 2xy + 1) + (x^2 + 2xy + y^2)$
$E = x^2 y^2 + x^2 + y^2 + 1$
Method: sos

$x^2 y^2 \geq 0$
$x^2 \geq 0$
$y^2 \geq 0$
Method: auto

$E \geq 1$
Method: sos

$E = 1$ when $x, y = 0$
Method: auto

Lemma expression_nonneg follows using method sos

Define theorem amc12a_2021_p7:
$\forall x, y \in \mathbb{R} : 1 \leq (xy - 1)^2 + (x + y)^2$
Proof:
Follows Lemma.
Method: sos, applying (auto simp:algebra_simps power2_eq_square) |

Figure 7: Typeset approximation of LEGO-Prover input lemmas (left) and found proof (right). See Figure 6 for the original Isabelle lemmas and proof. The proof proves that $\forall x, y \in \mathbb{R} : 1 \leq (xy - 1)^2 + (x + y)^2$. Skills 1 and 2 are the same; the fact that $x^2 \geq 0$ is used, though the exact proof differs from the lemmas. Skills 3 & 4 are also the same, though they do not seem to be used.

`357672c7751cd0c84aff6bf72a3d1bf97614e81d/data/full_data/valid` at the time of publication.

Note that the mean and standard deviation in Figure 1 are calculated using Python 3.8.9, numpy 1.22.2, `numpy.mean()` and `numpy.std()`.

Our experiments were run on an internal cluster, running one trial at a time. Each trial used 180 GB of RAM, 50 CPU cores, OpenAI credits, and ran within 24 hours. We upper bound the total compute time required to run our LEGO-Prover experiments at 96 hours. The full project required more compute than the experiments reported as one trial failed due to an out-of-memory error. Based on Wang et al. [2024a]'s estimate of \$300 per trial, we estimate the cost in OpenAI credits of our experiments to be \$7.38 per trial as we run half the number of attempts and one twentieth the number of questions. Under this estimate, the total cost of all our experiments is ~\$30.

Our code is modified from the released LEGO-Prover code base, available at `https://github.com/wiio12/LEGO-Prover` [Wang et al., 2024b], released under an MIT License. Evaluation is done using the miniF2F Zheng et al. [2022] dataset, available at `https://github.com/openai/miniF2F/tree/main`, which was released under the Apache License Version 2.0.

Our code is documented and released, alongside the generated LEGO-Prover logs. It is a minor modification to the existing code base, and there is no training stage or new limitations. The code is released under the same license as the parent repository.

## F TroVE Hyperparameters and Experiment Details

TroVE uses `CodeLlama-7b-Instruct-hf` [Rozière et al., 2023] interacting with the Python3 interpreter. We use the hyperparameters specified in the paper, outlined in Table 5. The same hyperparameters are used for the ablation, and our reproduction of baseline TroVE.

The mean and standard deviation of our 5 experiment runs are reported in Table 2. They are calculated using Python 3.8.9, numpy 1.22.2, `numpy.mean()` and `numpy.std()`. The 2-sided t-test reported the

Table 5: TroVE hyperparameters

| Hyperparameter | value |
|---|---|
| Library trim frequency (trim_steps) | 500 |
| Solution execution timeout in seconds (exec_timeout) | 100 |
| top-p (top_p) | 0.95 |
| Samples per prompt (num_return_sequences) | 5 |
| Temperature (temperature) | 0.6 |
| Max decode length (max_new_tokens) | 512 |

same table is performed using the same version of Python, scipy 1.8.1, `scipy.stats.ttest_ind()`, with the settings `equal_var=False` and `alternative='less'`.

Our experiments were run on an internal cluster, running up to 4 trials at once. Each trial used 1 Nvidia A40 GPU, 64 GB of RAM, 16 CPU cores, and ran within 12 hours. Smaller datasets completed more quickly. We upper bound the total compute time required to run our TroVE experiments at 480 hours. The full project required more compute than the experiments reported as we also tried running TroVE with quantized CodeLlama, CodeLlama 13B and 70B, and GPT-4o-mini.

Our code is modified from the released TroVE code base, available at `https://github.com/zorazrw/trove` [Wang et al., 2024b], which was released under the CC-BY-SA-4.0 license. Evaluation is done using the MATH Hendrycks et al. [2021] dataset, available at `https://github.com/hendrycks/math`, which was released under an MIT License.

Our code is documented and released, alongside the generated TroVE logs. It is a minor modification to the existing code base, and there is no training stage or new limitations. The code is released under the same license as the parent repository.

## F.1 Additional TroVE experiments

We also ran baseline TroVE using the larger CodeLlama 13B model, and found similar results with very little direct function use. The key difference with the 7B model was that a single function was learned for the geometry split, but it was never reused in a correct solution.

We also attempted to run baseline TroVE using the 70B model, however we discarded the results as the LLM's ethical safeguards were frequently tripped (e.g., giving reasons such as "it is not appropriate or ethical to provide assistance with academic assignments or graded exercises").

