# OpenReview forum: "Library Learning Doesn’t: The Curious Case of the Single-Use “Library”"
_NeurIPS.cc/2024/Workshop/MATH-AI — MATH-AI 24_

### Official Review · Reviewer_wRtp · 2024-09-27

**Rating:** 6
**Confidence:** 3

**Review:**

# Summary

The paper examines the claims that "library learning largely improves performance" made respectively by two previous works, LEGO-Prover and TROVE as a math problem solver, and finds out:

1. Lemma/Tool reuse rarely occurs in solutions generated by these 2 prover/math problem solvers
2. There is little to no effect of ablating reuse on the performance of these 2 prover/math problem solvers

# Strengths

## Originality

The paper identifies a potentially unsound claim in prior works that was overlooked. It would have been great if the prior works had included their own ablation studies to attribute and explain the performance gain.

## Quality

Results of the ablation experiments are convincing.

## Clarity

The paper gives a friendly summary of LEGO-Prover and TROVE, and clear descriptions of the ablation experiments.

## Significance

The paper serves as a message to the community encouraging researchers to conduct thorough ablation studies on their own work. More importantly, as mentioned in the paper, the ablation studies are constrained by available resources (original log for TROVE, the set of the ablation test set, etc.), so it should be read, discussed, and even challenged by more people, especially authors of the two prior works.


# Weaknesses

## Quality

1. It would be better to include a comparison between LEGO-Prover/TROVE and the cases where tool re-usage is the key to performance gain. **In what ways are they different such that their attributions of performance gain differ? In what ways are they similar such that the original authors believed that tool reuse was the key factor?** Answers to these questions will help further researchers avoid certain pitfalls and design systems that better leverage library learning.

2. **Do the problems in MiniF2F and MATH really require reuse? What if they are just so diverse that previously discovered lemma/tool is rarely applicable?** While the answer to this question won't change the observation that the 2 prior works may have overstated the contribution of reuse in their tasks, it will provide the 2 prior works a fair assessment of the effectiveness of their library learning modules.  For example, if there are only 5% of the problems that can benefit from reuse, then a 4% reuse rate shouldn't be considered low.


# Questions

1. Following up on bullet point 2 of Weaknesses, here is a slightly beyond-the-scope, but more important question about the evaluation process of all library learning papers: **If not MiniF2F/MATH, what's a good evaluation benchmark for library learning systems? How can we ensure enough data diversity and enough occurrences of reuse at the same time?**

---

### Official Review · Reviewer_oZyA · 2024-10-07
**Important new findings and insights in library learning systems**

**Rating:** 8
**Confidence:** 4

**Review:**

Summary:
The paper studies two library learning systems and introduces new findings suggesting that function reuse is extremely infrequent on miniF2F and MATH. Through ablation experiments, the authors further suggest that the primary drivers of the observed performance gains are self-correction and self-consistency, rather than function reuse.

Strengths:
- New, important findings that could have a significant impact on future research.
- The ablation experiments provide valuable insights into what contributes to improved performance in these library-learning systems.

Opportunities for improvement:
- The authors could mention which underlying LLM was used for the ablations in the main part of the paper. They could also discuss the impact of the underlying LLM and conduct experiments with different model sizes to assess whether function reuse could be beneficial for certain models.

Final remark:
The paper presents strong, impactful findings with clear evidence that self-correction and self-consistency are key factors in performance gains, rather than function reuse. Given the importance of these results and the clear writing style, I opt for a rating of 8.

---

### Official Review · Reviewer_MDiv · 2024-10-07
**Very good paper provide a critical view on two prior work on library learning for math problem solving**

**Rating:** 8
**Confidence:** 3

**Review:**

The paper investigates the role of library reuse in two prior works, LEGO-prover for mini2f and TroVE for MATH, and found that there is actually very little library reuse in the logs or their reproduced runs. Furthermore, they show that they can ablate the model by removing the library reuse components and still get roughly equal or better performance in the case of TroVE. This demonstrates that the performance gains of the two prior works can be achieved without library reuse. The paper (1) provides concrete evidence that little library reuse occurs in the two prior works by examining the provided logs or reproduced runs, and (2) ablates the model while still achieving competitive performance, showing that the gain may not come from library reuse at all but the self-consistency and the helper function style or problem decomposition methods. This work provides a critical and important investigation into library learning for math problem-solving.
Typo correction: line 69 lemm -> lemma

---

### Decision · Program_Chairs · 2024-10-07

Accept